# Reconstruction of Asphalt Pavements with Crumb Rubber Modified Asphalt Mixture in Cold Region: Material Characterization, Construction, and Performance

**DOI:** 10.3390/ma16051874

**Published:** 2023-02-24

**Authors:** Dongzhao Jin, Dongdong Ge, Jiaqing Wang, Lance Malburg, Zhanping You

**Affiliations:** 1Department of Civil, Environmental, and Geospatial Engineering, Michigan Technological University, 1400 Townsend Drive, Houghton, MI 49931, USA; 2Dickinson County Road Commission, 1107 S. Milwaukee Ave, Iron Mountain, MI 49801, USA

**Keywords:** dry process rubberized asphalt mixture, dynamic modulus, dynamic shear rheometer, noise test, pavement M-E design

## Abstract

Dry-processed rubberized asphalt mixture has recently attracted a lot of attention as an alternative to conventional asphalt mixtures. Dry-processed rubberized asphalt pavement has improved the overall performance characteristics compared to the conventional asphalt road. The objective of this research is to demonstrate the reconstruction of rubberized asphalt pavement and evaluate the pavement performance of dry-processed rubberized asphalt mixture based on laboratory and field tests. The noise mitigation effect of dry-processed rubberized asphalt pavement was evaluated at the field construction sites. A prediction of pavement distresses and long-term performance was also conducted using mechanistic-empirical pavement design. In terms of experimental evaluation, the dynamic modulus was estimated using materials test system (MTS) equipment, the low-temperature crack resistance was characterized by the fracture energy from the indirect tensile strength test (IDT), and the asphalt aging was assessed with the rolling thin-film oven (RTFO) test and the pressure aging vessel (PAV) test. The rheology properties of asphalt were estimated by a dynamic shear rheometer (DSR). Based on the test results: (1) The dry-processed rubberized asphalt mixture presented better resistance to cracking, as the fracture energy was enhanced by 29–50% compared to that of conventional hot mix asphalt (HMA); and (2) the high-temperature anti-rutting performance of the rubberized pavement increased. The dynamic modulus increased up to 19%. The findings of the noise test showed that at different vehicle speeds, the rubberized asphalt pavement greatly reduced the noise level by 2–3 dB. The pavement M-E (mechanistic-empirical) design-predicted distress illustrated that the rubberized asphalt pavement could reduce the IRI, rutting, and bottom-up fatigue-cracking distress based on a comparison of prediction results. To sum up, the dry-processed rubber-modified asphalt pavement has better pavement performance compared to the conventional asphalt pavement.

## 1. Introduction

Due to an increase in the number of vehicles on the road, millions of tires are produced each year [1,2]. An effective way to get rid of waste tire rubber and avoid an expanding environmental threat is to recycle it into asphalt pavement [3,4,5]. Rubberized asphalt mixture in a wet and dry process as an alternative to asphalt pavement has recently received a lot of attention.

Numerous researchers have focused on using crumb rubber to enhance the pavement performance in wet processes. Kirk et al. [6] found that rubber-modified asphalt mixture could keep the same road performance by reducing asphalt layer thickness by 20–50%. Some researchers have studied in high-temperature conditions the performance of rubber-modified hot mix asphalt (HMA). Xu et al. [7] and Shen et al. [8] found that the increased stiffness and viscosity of asphalt binder incorporated with recycled rubber could enhance the high temperature performance of the road. Some researchers have studied the crack energy of rubberized asphalt pavement. Nunez et al. [9] found that asphalt binder with 15% rubber in rubber-modified asphalt pavement could reduce crack development by five times when compared with conventional HMA. Peralta et al. [10] stated that rubber particles in HMA enhance the fracture energy and flexibility of the asphalt. Other researchers have studied the fatigue performance of rubber-modified asphalt pavement. Huang et al. [11] and Xiao et al. [12] found that when rubber is incorporated into asphalt, it could improve the viscosity of the asphalt and increase the percent of asphalt binder used in HMA, and it could improve the fatigue performance as well.

A number of researchers have made contributions to the application of crumb rubber-modified asphalt pavement in the dry process. Some researchers [13,14,15] have focused on the high-temperature properties and found that adding the scrap rubber into the dry process asphalt pavement significantly improved the rutting resistance. Some researchers have also studied the cracking resistance of rubberized HMA in dry processes. Chen et al. [16] and Xie et al. [17] indicated that the crumb rubber mix with conventional asphalt pavement could increase the fracture energy and increase the resistance of thermal cracking. Other researchers have studied the fatigue performance of crumb rubber-modified asphalt pavement in the dry process. Fontes et al. [18], Wang et al. [19], and Kocak et al. [20] found that the addition of crumb rubber increased the elastic component of the HMA and the fatigue resistance. 

According to the previous study, the study of dry process rubber asphalt mixture performance in the cold and wet regions is limited, and the long-term performance of crumb rubber-modified asphalt pavement still needs more effort and study, especially for the mechanistic-empirical pavement distress prediction for crumb rubber-modified asphalt pavement. The purpose of this study was to demonstrate the reconstruction of crumb rubber-modified asphalt pavement and evaluate the high-temperature and low-temperature properties of dry process rubberized HMA and predict the 20 years’ of service life pavement distress based on the Pavement M-E design. The dynamic modulus test and indirect tensile strength test were conducted, and rheology characteristics were evaluated based on the experimental performance. Additionally, the noise level of the road performance was evaluated before and after the reconstruction.

## 2. Raw Materials and Construction Information

### 2.1. Aggregate Gradation and Mix Design

Loose materials were collected from construction on County Road 607 (aka Bass Lake Road, MI, USA). Rubberized asphalt pavement by dry process and conventional asphalt pavement (without crumb rubber but with the same gradation) are compared in this project. Based on the Michigan Department of Transportation (MDOT) specification, the mix design is based on the Superpave mix design procedure (AASHTO M 323) for both dry-processed rubberized asphalt mixture and conventional asphalt mixture. Details of the aggregate gradation of the HMA are shown in Figure 1. The aggregate types and gradation of leveling layer (19 mm-HMA) are shown in Figure 1a. The aggregate types and gradation of the surface layer (12.5 mm-HMA) are displayed in Figure 1b. The (PG 58-34) bitumen was employed, and all mixtures used the same type of binder. The 19 mm-HMA and 12.5 mm-HMA asphalt mixtures’ design asphalt binder contents are 4.2 percent and 5.9 percent, respectively. The asphalt binder’s basic qualities meet all of the specification’s requirements. The mix temperature of dry process rubberized asphalt pavement is 163 °C. The reclaimed asphalt pavement (RAP) percentages of the 12.5 mm-HMA and 19 mm-HMA asphalt mixtures are 17% and 25%, respectively.

#### Crumb Rubber Materials

In this project, the crumb rubber was employed, as shown in Figure 2. The rubber used is 10% by weight of asphalt binder in this study. The surface of the broken rubber particle was treated by chemical engineering to make the rubber-modified asphalt more workable. This study used an anti-stripping agent to enhance the bond strength between aggregate and asphalt. It showed that the moisture sensitivity of the HMA satisfies the specification requirement. Anti-stripping agent dosages of 0.125 percent and 0.08 percent (by weight of binder) were utilized for 19 mm-HMA and 12.5 mm-HMA, respectively. It was applied in both crumb rubber-modified asphalt pavement and conventional asphalt pavement in this study. The basic properties and size distribution of the crumb rubber are shown in Table 1 and Table 2, respectively.

### 2.2. Preparation of Dry-Processed Rubberized Asphalt Mixture in the Plant

Figure 3 depicts the main elements of the plant. Cold feed bins are used to temporarily store aggregates before they are released in precise amounts onto the main conveyor as specified by the job mix formula (JMF). The aggregate mix is transported to the drum and heated with the other added components, such as bitumen and crumb rubber. In order to maintain the proper ratios of crumb rubber for all rates of production and batch sizes, the asphaltic cement pump at the asphalt plant interlocks with the ground tire rubber (GTR) feeder system. Finally, before being transported to the paving site, the prepared rubber-modified hot mix asphalt is kept in silos.

### 2.3. Selection of Pavement M-E Inputs

Traffic input has a great influence on the prediction of road performance. The road construction and thickness are affected by the average annual daily truck traffic (AADTT). Michigan’s low and medium traffic volume was selected. The typical lane’s design life is estimated to be 20 years. The pavement has two lanes with a speed limit of 65 miles per hour (105 km per hour). The truck volume is 51% in this study. In the design lane, 92 percent of the trucks have a medium traffic level, while 65% have a high traffic level. The growth rate of traffic is 2% (compound). The vehicle class distribution, monthly and hourly adjustment factors, and axle load distributions are based on the results from the Michigan Dept. of Transportation’s report [22].

### 2.4. Research Methodology

The following are the results of the laboratory mixture experiments: the dynamic modulus (|E*|), the indirect tensile strength of the HMA layer, and the complex shear modulus (|G*|) of the asphalt binder. Then, a sensitivity analysis of the DSR results and dynamic modulus results was estimated by M-E design based on the prediction of pavement distress. Figure 4 depicts the technical flowchart, including the following details.

### 2.5. Experiment Design

#### 2.5.1. Dynamic Modulus

Stone, sand, viscous asphalt binder, and additives make up the crumb rubber-modified asphalt mixture, which is a viscoelastic composite material. In the mechanical-empirical pavement design program, dynamic modulus (|E*|) is a critical material attribute. The loose mixture was compacted to 7 percent air void content. The master curve for |E*| was created according to the test results. The test was conducted in accordance with [23]. The approach established in 1999 was used to determine the creep compliance from the |E*| master curve [24].

#### 2.5.2. Indirect Tensile Strength (IDT)

The IDT strength and failure energy of the field condition obtained from the field core can be reflected using indirect tensile strength. The samples are conditioned for 3 h at −10 °C before the IDT test. The loading speed is 12.5 mm per minute. For the test, at least three field cores should be used. The load and displacement curves are used to compute the failure energy. The test is according to standard AASHTO T 322 [25]. The failure energy of an asphalt mixture is represented by equation (1) below.:(1)$$G_{f}=\frac{W_{f}}{D\times t}\times 10^{6}$$
where:

*G_f_* = failure energy (Joules/m^2^),

*W_f_* = work of failure (Joules),

*D* = specimen diameter (mm),

*t* = specimen thickness (mm).

#### 2.5.3. Dynamic Shear Rheometer (DSR)

The DSR test evaluates the viscosity and elasticity of asphalt binder. DSR is tested at temperatures ranging from 34 to 82 °C, with a 6 °C interval. Glass bottles shaped like cylinders are filled with all of the unaged asphalt, which is then placed within a spinning carriage inside of an oven for 85 min pass, with the carriage turning in the 163 °C oven. The short-term aging process of the samples is according to the rolling thin-film oven (RTFO) procedure. The pressure aging vessel (PAV) simulates a long-term aged asphalt. In the Pavement M-E program, the laboratory test results are used as M-E inputs, and the properties of the mixture are described as a level 1 input. The test results are based on at least three samples, and the DSR results are used to create the complex shear modulus (|G*|) master curve. The test is conducted according to AASHTO T 240 [26] and AASHTO R28 [27].

#### 2.5.4. Pavement M-E Analysis

The Pavement M-E was applied to determine the pavement performance, particularly considering the cracking and rutting distress [28]. The rate of vehicle increase was 2%, the pavement service life was designed as 20 years, the annual average daily truck traffic (AADTT) used in this study was 1000 and 2000, and the traffic volume design function was compound. The climate was selected in Dickinson County from the modern era retrospective-analysis for research and applications (MERRA) climate data for Pavement M-E Inputs. The calibration factor was determined using the instruction of Pavement M-E for MDOT, and the road’s pavement structure and thickness are provided in Table 3.

#### 2.5.5. The Noise Reduction Evaluation of Rubberized Asphalt Pavement with the Dry Process

The noise of asphalt pavements was assessed in this study, it has a 0.1 dB resolution and a 1.5 dB accuracy and it was 15 feet away from the car when the measurement was being made. Different roads’ noise levels were measured, as displayed in Figure 5. The Dodge Grand Caravan used for the noise measurement underwent noise testing at five different speeds.: 20 mph (32 km/h), 30 mph (48 km/h), 40 mph (64 km/h), 50 mph (80 km/h), and 60 mph (96 km/h). At each speed, a total of four noise measurements were taken. The ambient noise level was 46 dB, the temperature was 15 ºC at the time of the measurement.

## 3. Results and Discussions

### 3.1. Complex Shear Modulus (|G*|)

Phase angle and G* show the viscous and elastic properties of asphalt binders [29]. The results of G* for different types of asphalt binder are shown in Figure 6a. It can be determined that the rubber-modified asphalt shows a higher G*. This means the rubber increases the G* of asphalt binder at all of the frequencies and temperatures. Taking 12.5 mm-HMA and 12.5 mm-Rubber- HMA as examples, the complex shear modulus of 12.5 mm-HMA at 7.7 Hz is 18,504 Pa, and it increased to 20,331 Pa in the 12.5 mm-Rubber-HMA. Meanwhile, the higher reduced frequency shows a higher complex shear modulus because asphalt has more elastic components at high frequencies or low temperatures. 19 mm-HMA has a higher complex shear modulus compared to the 12.5 mm-HMA because the RAP usage in the leveling course is larger, and therefore, it increases the elastic component in the leveling layer asphalt. The results of the G* test on asphalt binder indicated that, when tested under different frequencies and temperatures, the rubber-modified asphalt binder exhibited greater stiffness in comparison to the conventional asphalt binder.

Figure 6b shows the phase angle test data from the DSR test. The phase angle decreased as the reduced frequency increased from the low value to the high value, which means the asphalt has a less viscous component at high frequency or low temperature. In addition, asphalt binder incorporated with rubber particles shows a lower phase angle compared to the base asphalt at all reduced frequencies. This illustrates that rubber incorporated with asphalt would exhibit an increased elastic component.

The rutting parameter (|G*|/sinδ) and fatigue parameter (|G*|×sinδ) can be used to reflect the rutting resistance and fatigue resistance of asphalt binder. The results of the rutting parameters and fatigue parameters of the asphalt binder are illustrated in Figure 7. The deformation resistance performance could be reflected by the rutting parameter. The 12.5 mm-HMA asphalt shows a higher rutting parameter than 19 mm-HMA asphalt, and the rubberized asphalt exhibits an increased rutting parameter than the control asphalt. For instance, the rutting parameter for 12.5 mm-HMA, 12.5 mm-rubber-HMA, 19 mm-HMA, and 19 mm-rubber-HMA at 58 °C is 23,326 Pa, 26,385 Pa, 15,710 Pa, and 28,406 Pa, respectively. This is because the rubber and reclaimed asphalt pavement (RAP) materials in 12.5 mm-HMA asphalt increase the elastic component and improve the resistance to permanent deformation. The fatigue parameter could be used to reflect fatigue cracking prevention. The viscous component of the complex shear modulus (|G*|×sinδ) should be minimized to dissipate energy through rebounding instead of cracking. Rubber-modified asphalt binder shows a lower fatigue parameter, which indicates that the rubber particles could enhance the fatigue cracking resistance. For example, the fatigue parameter for 12.5 mm-HMA, 12.5 mm-rubber-HMA, 19 mm-HMA, and 19 mm-rubber-HMA at 19 °C is 4.72 MPa, 2.20 MPa, 5.18 MPa, and 2.98 MPa, respectively. 19 mm-HMA has higher fatigue parameters compared to 12.5 mm-HMA. Because the leveling layer’s RAP content is higher, it may reduce the viscous component in asphalt extracted from the leveling layer’s loose materials.

### 3.2. Dynamic Modulus

The dynamic modulus could reveal the relationship between deformation and load [30]. Figure 8 illustrates the dynamic modulus for various kinds of asphalt mixtures. The dynamic modulus of the 12.5 mm rubber-HMA and 19 mm rubber-HMA are higher than those of the 12.5 mm-HMA and 19 mm-HMA. Rubber particles added to the HMA would improve the stiffness of the HMA. Meanwhile, the 19 mm-rubber-HMA and 19 mm-HMA have a higher dynamic modulus compared to 12.5 mm-rubber-HMA and 12.5 mm-HMA, respectively. The main reason behind this is that the RAP in the leveling layer is larger than that in the overlay course. The asphalt mixture E* test results showed that rubber-modified asphalt mixture showed higher stiffness compared to the conventional asphalt mixture at different frequencies and temperatures.

The stiffness of the HMA was estimated by the rutting parameter. Figure 9 illustrates the rutting parameter (|E*|/sinδ) at 10 Hz of various types of HMA. Rubber serves a crucial function in HMA. The rutting parameters of the rubberized HMA are of a larger value compared to the conventional hot mix. This means that the rubberized HMA has higher rutting resistance when compared to the control HMA. Furthermore, the RAP materials could increase the elastic content of the asphalt, so it may strengthen the rutting resistance of asphalt. Therefore, the leveling layer has higher rutting resistance than the surface layer. The findings suggest that the combination of a rubber-modified asphalt overlay and a rubber-modified asphalt leveling layer offers the most effective resistance against rutting, as opposed to the rubber-modified asphalt overlay with a conventional asphalt leveling layer or a conventional asphalt overlay with a rubber-modified asphalt leveling layer. Additionally, it can be inferred that the worst rutting performance can be anticipated from a conventional asphalt overlay with a conventional asphalt leveling layer.

### 3.3. IDT Strength and Failure Energy

The failure energy could be used to indicate the cracking potential [31]. The cracking properties of the asphalt mixture are reflected according to the failure energy. High failure energy means high cracking resistance. Figure 10a indicates that 12.5 mm-HMA shows high indirect tensile strength compared to 12.5 mm-rubber-HMA. It means that soft rubber particles in HMA may cause the mixture to become less rigid. Meanwhile, the 19 mm-HMA shows a higher indirect tensile strength compared to 12.5 mm-HMA because the RAP materials in 19 mm-HMA are high and increase the stiffness of the HMA. Figure 10b illustrates the failure energy of various types of HMA. It is obvious that the rubberized HMA shows higher failure energy and better cracking resistance performance. Meanwhile, the 19 mm-HMA asphalt mixture shows lower failure energy compared to 12.5 mm-HMA, which means the RAP has a negative effect on cracking resistance. The lower peak load in rubber mixtures is mostly due to the increased viscosity of the modified binder, which causes a damping effect [32]. The higher failure energy at lower temperatures is due to the crack pinning effects of rubber particles that impede crack propagation [33]. The results indicated that the rubber-modified asphalt mixture showed better cracking resistance when compared with the conventional asphalt mixture. It indicates the rubber-modified asphalt overlay with rubber-modified asphalt leveling layer would provide the best cracking resistance compared to the rubber-modified asphalt overlay with conventional asphalt leveling layer and conventional asphalt overlay with rubber-modified asphalt leveling layer. And the conventional asphalt overlay with a conventional asphalt leveling layer is expected to show the worst cracking performance.

### 3.4. Pavement Distress Prediction Results

The predicted pavement distress results at 1000 AADTT level are shown in Figure 11a. From the IRI, Total rut, AC rut, B-U, and T-D cracking results, there are noticeable discrepancies in the cracking outcomes for the four different pavement structures. AC rut and total rut indicated that the rubberized asphalt pavement exhibits the best rutting resistance, and the rubber-modified overlay and rubber-modified leveling layer have the same rutting resistance, while all three structures have improved stiffness than control asphalt pavement. The total rutting resistance between rubber-modified asphalt pavement and conventional asphalt pavement increased by 34.6%. AC top-down cracking and AC bottom-up results show the cracking resistance of asphalt pavement. This means the rubberized asphalt pavement has enhanced cracking resistance. Meanwhile, the rubber-modified overlay has superior cracking resistance when compared with the rubber-modified leveling layer, and all of the rubber-modified asphalt pavement shows better cracking performance compared with conventional asphalt pavement. IRI shows an increase in roughness, which is caused by the occurrence of surface distress on the pavement. Therefore, conventional asphalt pavement shows a higher IRI value compared to rubberized asphalt pavement. The predicted rutting and cracking results are in agreement with the dynamic modulus and fracture properties experimental test results. It is worth mentioning that the asphalt binder properties used in this study were extracted using the standard solvent extraction method, which could not extract all the rubber particles present in the loose HMA; therefore, the prediction distress of the rubber-modified HMA could not be fully reflected in the Pavement M-E analysis.

Figure 11b illustrates the predicted pavement distress results at the 2000 AADTT level. It is noteworthy that in the total rut and AC rut results, the rubberized asphalt pavement has better permanent deformation performance compared to the conventional hot mix pavement. Meanwhile, AADTT increased from the 1000 to the 2000 level. Take the total rut as an example. The total rut of conventional asphalt pavement increased from 0.35 inches to 0.41 inches, while the total rut of rubberized asphalt pavement increased from 0.26 inches to 0.31 inches. AC bottom-up cracking and top-down cracking results show that compared to conventional hot mix pavement, the rubberized asphalt pavement has better cracking resistance. Moreover, this continued when the AADTT increased from the 1000 to the 2000 level. Now take AC bottom-up cracking as an example. The AC bottom-up cracking of conventional asphalt pavement increased from 12.63 to 14.71, while the AC bottom-up cracking of rubber-modified asphalt pavement increased from 12.26 to 14.23. Based on the above analysis, it is indicated that the rubber-modified asphalt pavement could increase the high temperature and low temperature performance with an increased traffic volume level on low traffic volume roads. The Pavement M-E input for 12.5 mm-HMA, 12.5 mm-rubber-HMA, 19 mm-HMA, and 19 mm-rubber-HMA are shown in Table 4, Table 5, Table 6 and Table 7.

## 4. Field Construction and Noise Level Measurement

The field construction process is shown in Figure 12. The old road distress is illustrated in Figure 12a, and it can be illustrated that moisture damage, fatigue cracking, and low temperature cracking appear on the road. The existing pavement was crushed back to gravel and mixed with the existing gravel road base. This recycled mix was graded and compacted, and then the pavement leveling layers were repaved, as shown in Figure 12b. After that, the leveling layer was covered by the asphalt emulsion, as shown in Figure 12c. The paving and compaction procedure of the overlay is displayed in Figure 12d,e, respectively. The pavement condition (after 2 years) is shown in Figure 12f–h. The pavement conditions (after 3 years) are shown in Figure 12i,j. The difference between rubber-modified asphalt pavement and the old pavement without rehabilitation can be obviously observed. It can be seen that the road condition is still good after two years of services.

This study measured the noise levels of three types of asphalt pavement test sections and old asphalt pavement. The noise results are displayed in Figure 13. The asphalt pavement noise increased linearly with the increase in vehicle speed. Compared to the old asphalt pavement, the noise was reduced by more than 5.5 dB under all vehicle speed conditions, and the noise was reduced by 7.1 dB at the speed of 60 mph (96 km/h). The noise energy of the newly paved asphalt pavement was reduced to less than 28.3% of the noise energy of the old asphalt pavement. At the speed of 60 mph (96 km/h), the noise energy is reduced to 19.5% of the noise energy of old asphalt pavement. Structure #2 and structure #4 were all rubberized asphalt mixtures, and the noise results of the two road sections were similar. Structure #1 was the conventional asphalt mixture. Compared with structure #2 and structure #4, the noise of structure #1 without rubber was higher. The noise of structure #2 and structure #4 was 2 dB lower than that of structure #1, except for the noise at the speed of 20 mph (32 km/h). At the design speed of 50 mph (80 km/h), the noise was reduced by more than 3 dB, and the noise energy was reduced by more than 50%. The field noise measurement of the sound level indicated that the rubber modification could unquestionably reduce the noise generated from the asphalt pavement and vehicle tires since compared with the conventional hot mix pavement, the noise energy was reduced by more than 50% at the design speed of 50 mph (80 km/h). The Pavement Surface Evaluation and Rating (PASER) data of the old pavement and new reconstruction pavement are displayed in Table 8. PASER value has three categories (8–10 is good condition, 5–7 is fair condition, 1–4 is poor condition). It can be seen that the PASER rating was only 2 in 2017, which means the original pavement was in very poor condition, and the PASER rating increased to 10 in 2019 after construction. The PASER rating was 8 in 2021, which means the pavement was still in very good condition without needing maintenance.

## 5. Summary and Conclusions

The objective of this research is to demonstrate the reconstruction of rubberized asphalt pavement and evaluate the pavement performance of dry-processed rubberized asphalt mixture based on laboratory and field tests. Some findings can be summarized below:(1)The dynamic modulus test showed that the rubber-modified asphalt mixture has higher rigidity compared to that of a typical asphalt mixture, in which the dynamic modulus increased up to 19% at a different reduced frequency. The improved dynamic modulus could contribute to a better rutting resistance at various temperatures and frequencies.(2)The indirect tensile test results showed that the rubberized asphalt mixture enhances the cracking resistance compared to the typical asphalt mixture at low temperatures since the failure energy was increased by 29–50% after rubber-modification compared to that of control HMA.(3)The evaluation results of the asphalt binder reveal that compared to conventional asphalt, the rubber-modified asphalt binder shows improved high-temperature and fatigue properties.(4)The noise results illustrated that the rubberized asphalt pavement significantly reduced the noise level by 2–3 dB on the road compared to newly constructed conventional asphalt pavement at various traffic speeds. The noise mitigation effect of rubber-modified asphalt pavement was verified in the field.(5)The pavement M-E analysis showed that the rubber-modified asphalt pavement could reduce the IRI, rutting, and bottom-up fatigue-cracking in comparison with that of the normal asphalt pavement.

In conclusion, the use of rubber asphalt mixes that have undergone dry processing during pavement construction increased the pavement’s resistance to permanent deformation, thermal cracking, and fatigue. Meanwhile, the tire noise was also obviously mitigated on rubber-modified asphalt pavement. Consequently, the dry-processed rubber-modified asphalt pavement has better performance than the conventional asphalt pavement.

## Figures and Tables

**Figure 1 materials-16-01874-f001:**
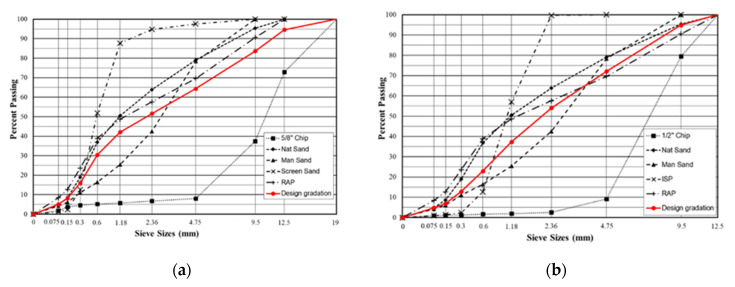
The specified asphalt mixes’ aggregate gradation. (**a**) The gradation design in leveling course; (**b**) the gradation design in surface course.

**Figure 2 materials-16-01874-f002:**
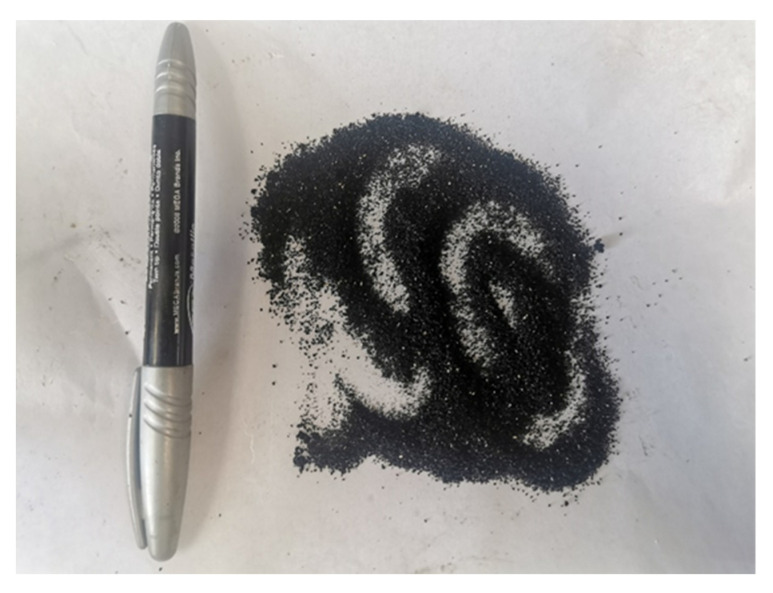
The crumb rubber used in this study.

**Figure 3 materials-16-01874-f003:**
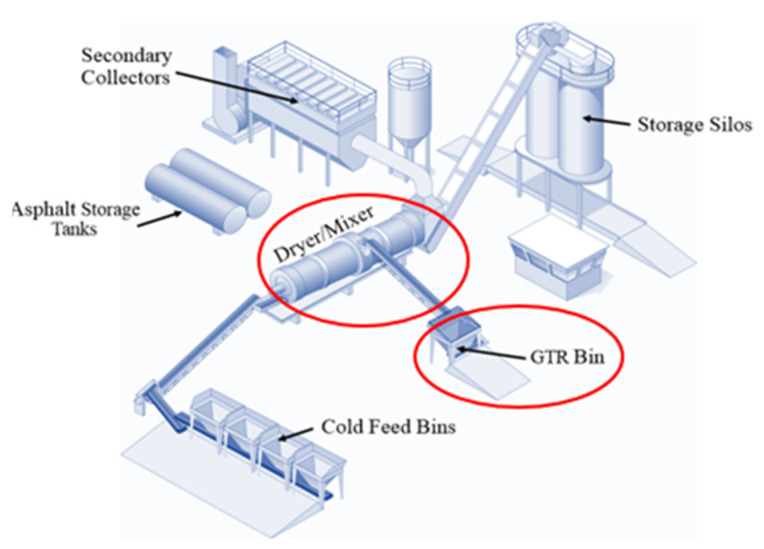
Demonstration project in Dickinson (Revised from Mamlouk [21]).

**Figure 4 materials-16-01874-f004:**
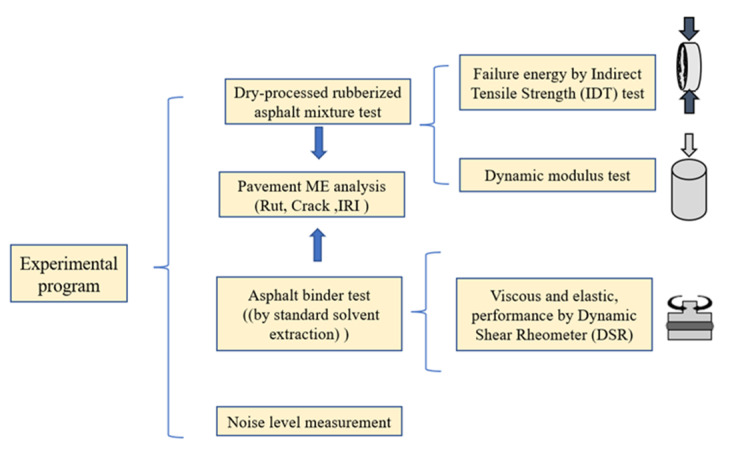
Technical flowchart used in the research.

**Figure 5 materials-16-01874-f005:**
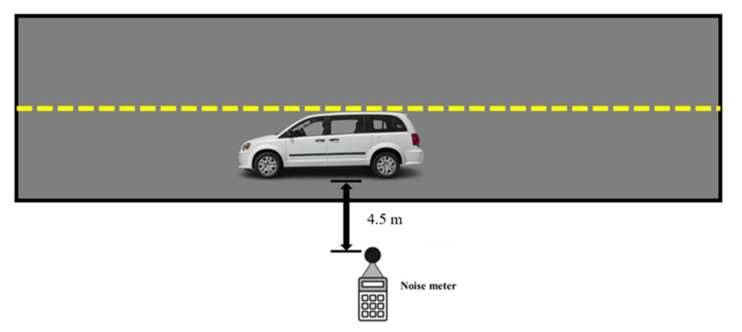
Demonstration of the noise measurement procedure.

**Figure 6 materials-16-01874-f006:**
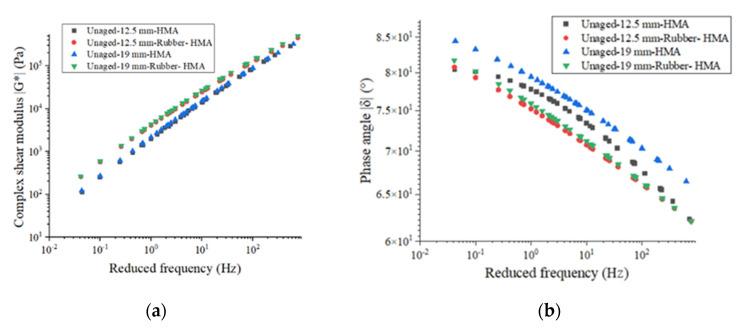
The master curves of the asphalt binder. (**a**) G* master curve @ 58 °C; (**b**) phase angle master curve @ 58 °C.

**Figure 7 materials-16-01874-f007:**
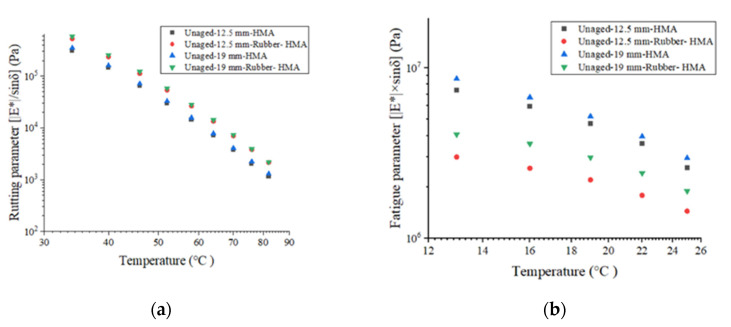
The asphalt binder potential of anti-rutting and anti-fatigue. (**a**) The rutting parameter (|G*|/sinδ) of asphalt binder at 10 rad/s; (**b**) the fatigue parameter (|G*|×sinδ) of asphalt binder at 10 rad/s.

**Figure 8 materials-16-01874-f008:**
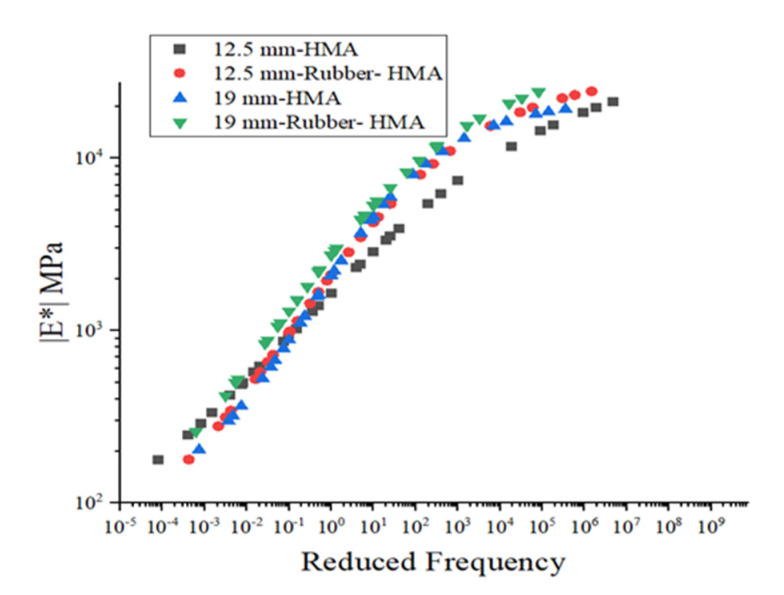
Dynamic modulus mastercurve of HMA (@ 21 °C).

**Figure 9 materials-16-01874-f009:**
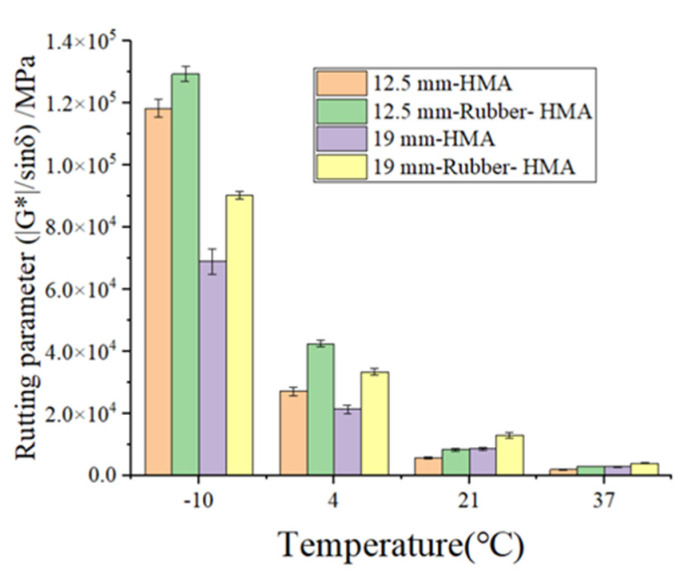
Rutting parameter (|G*|/sinδ) at 10 Hz of various types of HMA.

**Figure 10 materials-16-01874-f010:**
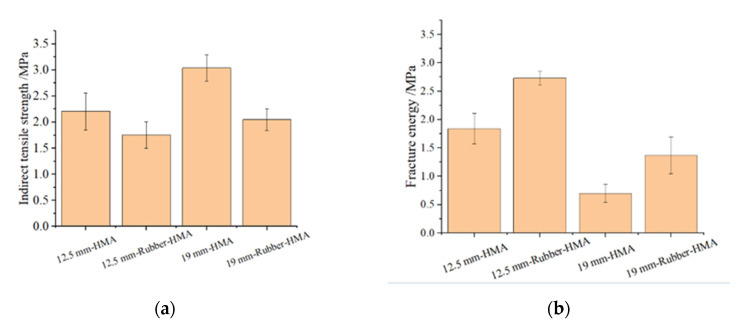
IDT strength and failure energy results. (**a**) IDT strength test data; (**b**) failure energy results.

**Figure 11 materials-16-01874-f011:**
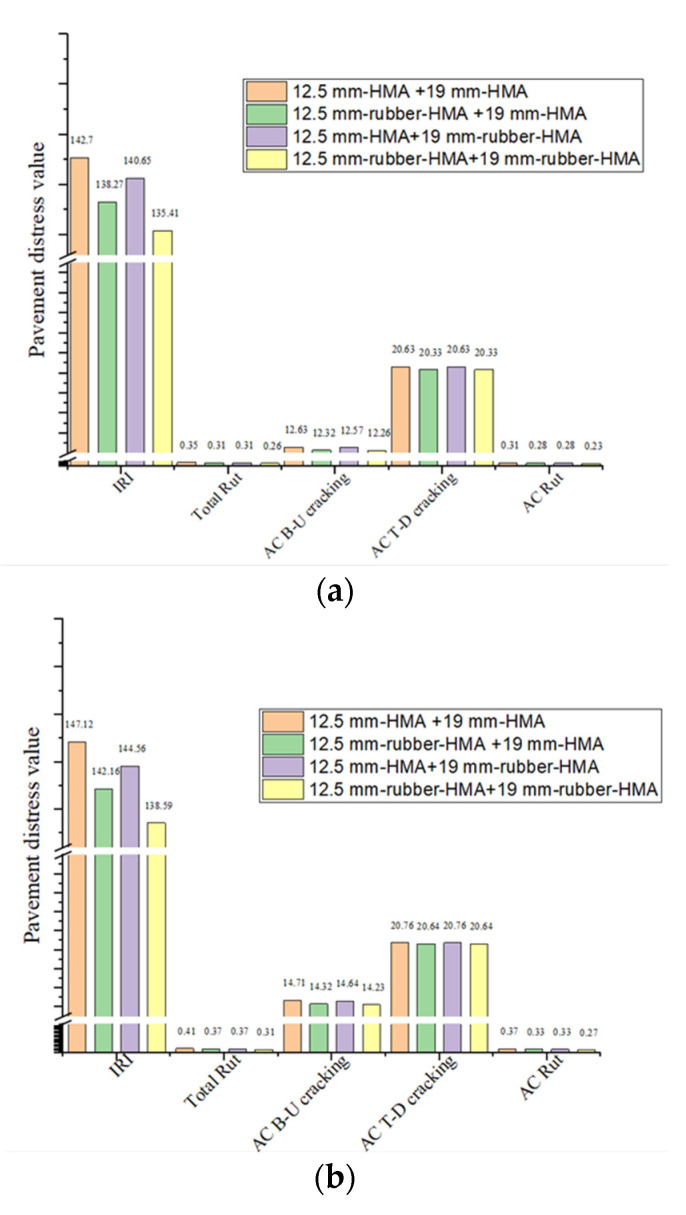
Results of pavement distress for several types of HMA. Note: international roughness index (IRI), AC (bottom-up) B-U cracking, AC (top-down) T-D cracking. (**a**) Pavement distress results (AADTT = 1000); (**b**) pavement distress results (AADTT = 2000).

**Figure 12 materials-16-01874-f012:**
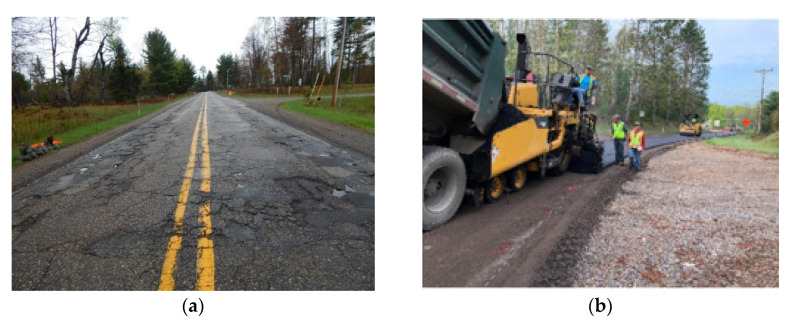
Field construction in Dickinson project. (**a**) Original pavement condition; (**b**) leveling layer compaction; (**c**) emulsion asphalt application for the leveling layer; (**d**) surface layer laydown; (**e**) surface layer paving and compaction; (**f**) rubber-modified asphalt pavement condition after construction (2 years); (**g**) rubber-modified overlay pavement condition after construction (2 years); (**h**) conventional pavement condition after construction (2 years); (**i**) old pavement without rehabilitation vs. pavement with rubber asphalt surface (3 years); (**j**) rubber-modified overlay pavement after construction (3 years).

**Figure 13 materials-16-01874-f013:**
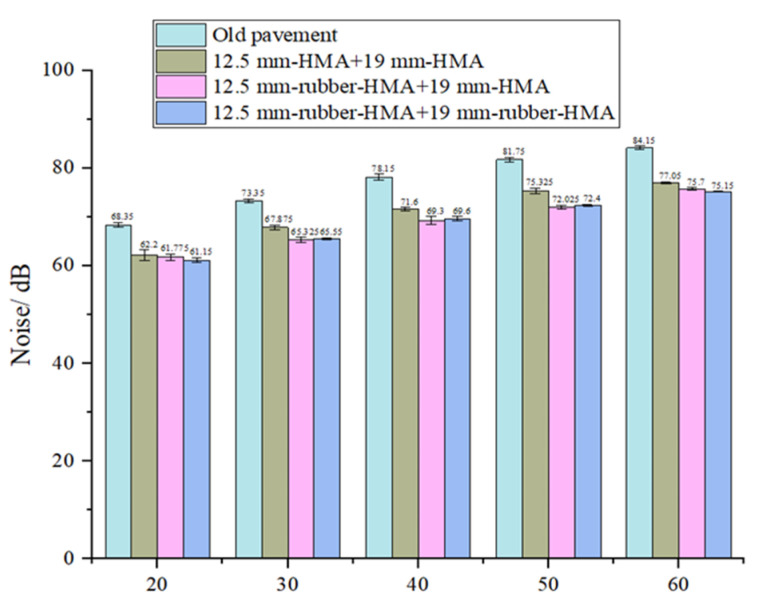
Noise test results for the old pavement section and pavement sections after construction.

**Table 1 materials-16-01874-t001:** Crumb rubber’ basic properties used in this study.

Properties	Results
Appearance	Black, fine grained scrap rubber with white flecks
Specific gravity	1.15
Flash point, ignition temperature	246 °C, 370 °C

**Table 2 materials-16-01874-t002:** Crumb rubber size and weight distribution used in this study.

Sieve	Passing %
No.16	100
No.30	96–99
No.40	70–74
No.100	42–48
No.200	0–12

**Table 3 materials-16-01874-t003:** Pavement materials and information used in this study.

Layer Types and Thickness	Structure-1	Structure-2	Structure-3	Structure-4
Surface layer (3.8 cm)	12.5 mm-HMA	12.5 mm—Rubber-HMA	12.5 mm-HMA	12.5 mm—Rubber-HMA
Leveling course (5 cm)	19 mm-HMA	19 mm-HMA	19 mm—Rubber-HMA	19 mm—Rubber-HMA
Asphalt base course (17.8 cm)	Dense asphalt pavement course			
Aggregate base (15.2 cm)	Sandwich granular			
Aggregate base (25.4 cm)	Crushed stone			
Aggregate subbase (15.2 cm)	Crushed gravel			
subgrade	Semi-infinite			

Note: Structure-3 was not used in the construction. The asphalt base layer, aggregate base, aggregate subbase, and subgrade value is based on the MDOT report results.

**Table 4 materials-16-01874-t004:** Pavement M-E input for 12.5 mm- HMA.

|E*| (MPa) Average Value							
	F (Hz)	0.1	0.5	1	5	10	25
T (°C)							
−10		10,332	14,152	15,911	18,107	19,640	21,548
10		2542	3980	4803	6767	7805	9205
21		678	1044	1270	2192	2686	3398
37		353	497	655	815	993	1570
54		176	248	328	408	496	785
**|G*| (MPa) Average Value**				**Creep Compliance**			
Temperature (°C)	Binder G* (Pa)		Phase angle (°)	Time (sec)	Temperature (°C)		
13	5,874,500		42.9		−20	−10	0
25	1,439,000		49.6	1	3.31 × 10^−7^	5.85 × 10^−7^	1.15 × 10^−6^
46	32,997		68.7	2	3.56 × 10^−7^	6.42 × 10^−7^	1.29 × 10^−6^
58	7005		72.8	5	3.90 × 10^−7^	7.21 × 10^−7^	1.49 × 10^−6^
82	476.8		81.7	10	4.19 × 10^−7^	7.99 × 10^−7^	1.69 × 10^−6^
(−10°C) IDT strength: 2.2 MPa				20	4.55 × 10^−7^	8.85 × 10^−7^	1.91 × 10^−6^
				50	5.05 × 10^−7^	1.01 × 10^−7^	2.23 × 10^−6^
				100	5.47 × 10^−7^	1.13 × 10^−7^	2.56 × 10^−6^

**Table 5 materials-16-01874-t005:** Pavement M-E input for 19 mm- HMA.

|E*| (MPa) Average Value							
	F (Hz)	0.1	0.5	1	5	10	25
T (°C)							
−10		10,333	13,319	15,029	18,046	19,801	20,763
10		3788	5663	6665	8801	10,203	11,007
21		874	1499	1941	3339	4260	5145
37		406	559	672	1234	1579	2222
54		203	280	336	617	789	1111
**|G*| (MPa) Average Value**				**Creep Compliance**			
Temperature (°C)	Binder G* (Pa)		Phase angle (°)	Time (sec)	Temperature (°C)		
13	5,542,000		43.4		−20	−10	0
25	1,392,400		50	1	3.94 × 10^−7^	6.29 × 10^−7^	1.17 × 10^−6^
46	32,135		69	2	4.24 × 10^−7^	7.08 × 10^−7^	1.37 × 10^−6^
58	6596		73.5	5	4.76 × 10^−7^	8.31 × 10^−7^	1.74 × 10^−6^
82	435		82.3	10	5.20 × 10^−7^	9.60 × 10^−7^	2.09 × 10^−6^
(−10°C) IDT strength: 3.04 MPa				20	5.72 × 10^−7^	1.12 × 10^−6^	2.53 × 10^−6^
				50	6.66 × 10^−7^	1.37 × 10^−6^	3.38 × 10^−6^
				100	7.49 × 10^−7^	1.63 × 10^−6^	4.20 × 10^−6^

**Table 6 materials-16-01874-t006:** Pavement M-E input for 12.5 mm-rubber-HMA.

|E*| (MPa) Average Value							
	F (Hz)	0.1	0.5	1	5	10	25
T (°C)							
−10		12,656	16,536	18,284	21,951	23,919	25,259
10		4939	6982	8035	10,964	12,338	13,826
21		898	1511	1927	3320	3930	4750
37		360	512	664	1253	1604	1949
54		180	256	332	626	802	975
**|G*| (MPa) Average Value**				**Creep Compliance**			
Temperature (°C)	Binder G* (Pa)		Phase angle (°)	Time (sec)	Temperature (°C)		
13	2,529,100		42.6		−20	−10	0
25	1,148,300		47.7	1	3.08 × 10^−7^	5.19 × 10^−7^	1.03 × 10^−6^
46	43,641		66.1	2	3.34 × 10^−7^	5.81 × 10^−7^	1.20 × 10^−6^
58	9633		70.3	5	3.70 × 10^−7^	6.88 × 10^−7^	1.49 × 10^−6^
82	661.6		79.48	10	4.04 × 10^−7^	7.80 × 10^−7^	1.79 × 10^−6^
(−10°C) IDT strength: 1.76 MPa				20	4.47 × 10^−7^	8.97 × 10^−7^	2.15 × 10^−6^
				50	5.10 × 10^−7^	1.10 × 10^−6^	2.77 × 10^−6^
				100	5.69 × 10^−7^	1.28 × 10^−6^	3.44 × 10^−6^

**Table 7 materials-16-01874-t007:** Pavement M-E input for 19 mm-rubber-HMA.

|E*| (MPa) Average Value							
	F (Hz)	0.1	0.5	1	5	10	25
T (°C)							
−10		11,791	15,328	16,938	20,778	22,689	23,055
10		4180	6087	6984	9567	10,668	11,766
21		919	1640	2100	3827	4725	6222
37		521	831	1059	1888	2316	2969
54		261	416	529	944	1158	1485
**|G*| (MPa) Average Value**				**Creep Compliance**			
Temperature (°C)	Binder G* (Pa)		Phase angle (°)	Time (sec)	Temperature (°C)		
13	2,607,500		41.3		−20	−10	0
25	1,370,800		47.5	1	3.38 × 10^−7^	5.69 × 10^−7^	1.07 × 10^−6^
46	51,869		65.6	2	3.66 × 10^−7^	6.41 × 10^−7^	1.26 × 10^−6^
58	10,588		70.45	5	4.14 × 10^−7^	7.58 × 10^−7^	1.60 × 10^−6^
82	690.1		79.8	10	4.55 × 10^−7^	8.80 × 10^−7^	1.90 × 10^−6^
(−10°C) IDT strength: 2.02 MPa				20	5.04 × 10^−7^	1.02 × 10^−6^	2.30 × 10^−6^
				50	5.91 × 10^−7^	1.25 × 10^−6^	3.03 × 10^−6^
				100	6.65 × 10^−7^	1.50 × 10^−6^	3.71 × 10^−6^

**Table 8 materials-16-01874-t008:** PASER rating data of the old pavement and new reconstruction pavement.

	Year	2013	2015	2017		Year	2019	2021
Pavement Type					Reconstruction Pavement Type			
Control section #1		4	3	2	12.5 mm-HMA + 19 mm-HMA section		10	8
Control section #2		2	2	2	12.5 mm-rubber-HMA + 19 mm-HMA		10	8
Control section #3		2	2	2	12.5 mm-rubber-HMA + 19 mm-rubber-HMA		10	8

Note: Control section #1 is reconstructed as 12.5 mm-HMA + 19 mm-HMA section, Control section #2 is reconstructed as 12.5 mm-rubber-HMA + 19 mm-HMA, Control section #3 is reconstructed as 12.5 mm-rubber-HMA + 19 mm-rubber-HMA.

## Data Availability

On reasonable request, the corresponding author will make the datasets generated during the current investigation available.

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
