# Peer review of "Reconstruction of Asphalt Pavements with Crumb Rubber Modified Asphalt Mixture in Cold Region: Material Characterization, Construction, and Performance"

_materials, 2023, doi:10.3390/ma16051874_

Round 1
Reviewer 1 Report
Dry-processed rubberized asphalt mixture and pavement were studied in this manuscript, and found dry-processed rubberized asphalt would improve the overall performance characteristics compared with a conventional asphalt road, which shows better resistance to cracking, higher fracture energy, better high-temperature antiutting performance and lower noise level.
Asphalt mixture is a kind of engineering material with important application in road construction, which is of great significance both in theory and application. In this manuscript, the existing asphalt aggregate is improved on the process of dry-processed rubberized asphalt mixture and show good effect, has a good reference value, so it is recommended to be published after major revision as following:
1. The research content is not deep enough, so it is suggested to supplement the structural difference between dry-processed rubberized asphalt mixture and traditional asphalt mixture, and elaborate the structural basis of the excellent properties of the new process and material.
2 Dry-processed rubberized asphalt is the focus of this paper, but the specific procedure, rubber varieties, modification methods and so on that may have an important impact on the research content were not described clearly.
Author Response
Dear Editors and Reviewers,
We thank you very much for giving us an opportunity to revise our manuscript and we have seriously taken into account these modification proposals. The explanations of what we have changed in response to the comments are also given point by point in the following pages attached in this paper. The corresponding changes have been marked in red background in the revised manuscript, respectively. The grammar and language issues of the whole manuscript have been checked carefully. More details could be found in the attachment
We hope that all these changes fulfill the requirements to make the manuscript acceptable for publication.
Looking forward to hearing from you.
Sincerely yours,
Dongzhao Jin, Dongdong Ge, Jiaqing Wang, Lance Malburg, Zhanping You

Reviewer 2 Report
This paper is meaningful for engineering practice, and the methods and results are clearly presented. Main comments are listed as the following:
1. The introduction needs a significant improvement. The state of the art and the main limitation on the research topic are suggested to be clearly presented.
2. More information on the crumb rubber (like the mesh number) and chemical additives used is suggested to be given.
3. The full name of PMED should be given in the main body of the manuscript.
4. Section 3.1: the complex shear modulus G* is a complex quantity composed of both real part (storage modulus) and imaginary part (loss modulus), and it should be distinguished from the dynamic shear modulus |G*|, which is a real quantity.
5. Fig. 9: |E*| should be |G*|.
6. The objective should be consistent in both abstract and summary.
Author Response
Dear Editors and Reviewers,
We thank you very much for giving us an opportunity to revise our manuscript and we have seriously taken into account these modification proposals. The explanations of what we have changed in response to the comments are also given point by point in the following pages attached in this paper. The corresponding changes have been marked in the red background in the revised manuscript, respectively. The grammar and language issues of the whole manuscript have been checked carefully.
We hope that all these changes fulfill the requirements to make the manuscript acceptable for publication.
Looking forward to hearing from you.
Sincerely yours,
Dongzhao Jin, Dongdong Ge, Jiaqing Wang, Lance Malburg, Zhanping You

Reviewer 3 Report
This study presents various data on the experiment and construction of asphalt pavement using crumb rubber modified asphalt mixture. However, the composition of the manuscript is more like a report format than a specialized journal and the novelty is not high, so major revisions and supplements must be made.
Others)
- The purpose of the study is unclear.
- Too much space is devoted to the explanation of the test method.
- Lack of differentiation from previous studies.
- It is not necessary to include all construction photos. Include only what's important.
- Lack of logical basis for results.
-The style such as size and line spacing is also very different from the journal's format.
Author Response
Dear Editors and Reviewers,
We thank you very much for giving us an opportunity to revise our manuscript and we have seriously taken into account these modification proposals. The explanations of what we have changed in response to the comments are also given point by point in the following pages attached in this paper. The corresponding changes have been marked in red background in the revised manuscript, respectively. The grammar and language issues of the whole manuscript have been checked carefully.
We hope that all these changes fulfill the requirements to make the manuscript acceptable for publication.
Looking forward to hearing from you.
Sincerely yours,
Dongzhao Jin, Dongdong Ge, Jiaqing Wang, Lance Malburg, Zhanping You

Round 2
Reviewer 1 Report
The manuscript discusses the influence of road material processing on the performance, which has a certain reference value. Compared with the original draft, the revised version supplemented the description of the materials, which improved the paper to a certain extent. However, it did not involve the relationship among the processing,structure andperformance of the material, so the paper shows some superficial. It is suggested to pay more attention to the relationship among the processing,structure andperformance of the material in discussion.
Author Response
Dear Editors and Reviewers,
We thank you very much for giving us an opportunity to revise our manuscript and we have seriously taken into account these modification proposals. The explanations of what we have changed in response to the comments are also given point by point in the following pages attached in this paper. The corresponding changes have been marked in red background in the revised manuscript, respectively. The grammar and language issues of the whole manuscript have been checked carefully.
We hope that all these changes fulfill the requirements to make the manuscript acceptable for publication.
Looking forward to hearing from you.
Sincerely yours,
Dongzhao Jin, Dongdong Ge, Jiaqing Wang, Lance Malburg, Zhanping You
Responses to reviewer’s comments are listed as follows:
(Corresponding changes have been marked in red in the revised manuscript)
=========================================
Review 1
The manuscript discusses the influence of road material processing on the performance, which has a certain reference value. Compared with the original draft, the revised version supplemented the description of the materials, which improved the paper to a certain extent. However, it did not involve the relationship among the processing,structure and performance of the material, so the paper shows some superficial. It is suggested to pay more attention to the relationship among the processing,structure and performance of the material in discussion.
Comment #1: The manuscript discusses the influence of road material processing on the performance, which has a certain reference value. Compared with the original draft, the revised version supplemented the description of the materials, which improved the paper to a certain extent. However, it did not involve the relationship among the processing,structure and performance of the material, so the paper shows some superficial. It is suggested to pay more attention to the relationship among the processing,structure and performance of the material in discussion.
Response: Thank you so much for your comments, lots of revision has been done as follows:
- The sentence has been added into Line 208-210 “The results of the G* test on asphalt binder indicated that, when tested under different frequencies and temperatures, the rubber-modified asphalt binder exhibited greater stiffness in comparison to the conventional asphalt binder.”
- The sentence has been added into Line 246-248 “The asphalt mixture E* test results showed that rubber modified asphalt mixture showed higher stiffness compared with the conventional asphalt mixture at different frequency and temperature”.
- The sentence has been added into Line 254-259 “The findings suggest that the combination of a rubber-modified asphalt overlay and a rub-ber-modified asphalt leveling layer offers the most effective resistance against rutting, as op-posed to the rubber-modified asphalt overlay with a conventional asphalt leveling layer or a conventional asphalt overlay with a rubber-modified asphalt leveling layer. Additionally, it can be inferred that the worst rutting performance can be anticipated from a conventional asphalt overlay with a conventional asphalt leveling layer.”
- The sentence has been added into Line 275-281 “The results indicated that the rubber-modified asphalt mixture showed better cracking resistance when compared with the conventional asphalt mixture. It indicates the rubber-modified asphalt overlay with rubber-modified asphalt leveling layer would provide the best cracking resistance compared with the rubber-modified asphalt overlay with conventional asphalt leveling layer and conventional asphalt overlay with rubber-modified asphalt leveling layer. And the conven-tional asphalt overlay with conventional asphalt leveling layer would be expected the worst cracking performance. “
- The sentence has been added into Line 299-300 “The predicted rutting and cracking results are in agreement with the dynamic modulus and fracture properties experimental test results.”

Reviewer 3 Report
It was judged that the authors appropriately supplemented the manuscript by reflecting the review opinions.
Author Response
Dear Editors and Reviewers,
We thank you very much for giving us an opportunity to revise our manuscript and we have seriously taken into account these modification proposals. The explanations of what we have changed in response to the comments are also given point by point in the following pages attached in this paper. The corresponding changes have been marked in red background in the revised manuscript, respectively. The grammar and language issues of the whole manuscript have been checked carefully.
We hope that all these changes fulfill the requirements to make the manuscript acceptable for publication.
Looking forward to hearing from you.
Sincerely yours,
Dongzhao Jin, Dongdong Ge, Jiaqing Wang, Lance Malburg, Zhanping You
Responses to reviewer’s comments are listed as follows:
(Corresponding changes have been marked in red in the revised manuscript)
=========================================
Review 3
It was judged that the authors appropriately supplemented the manuscript by reflecting the review opinions.
Comment #1: It was judged that the authors appropriately supplemented the manuscript by reflecting the review opinions.
Response: Thank you so much for your comments, lots of revision has been done as follows:
- The sentence has been added into Line 208-210 “The results of the G* test on asphalt binder indicated that, when tested under different frequencies and temperatures, the rubber-modified asphalt binder exhibited greater stiffness in comparison to the conventional asphalt binder.”
- The sentence has been added into Line 246-248 “The asphalt mixture E* test results showed that rubber modified asphalt mixture showed higher stiffness compared with the conventional asphalt mixture at different frequency and temperature”.
- The sentence has been added into Line 254-259 “The findings suggest that the combination of a rubber-modified asphalt overlay and a rub-ber-modified asphalt leveling layer offers the most effective resistance against rutting, as op-posed to the rubber-modified asphalt overlay with a conventional asphalt leveling layer or a conventional asphalt overlay with a rubber-modified asphalt leveling layer. Additionally, it can be inferred that the worst rutting performance can be anticipated from a conventional asphalt overlay with a conventional asphalt leveling layer.”
- The sentence has been added into Line 275-281 “The results indicated that the rubber-modified asphalt mixture showed better cracking resistance when compared with the conventional asphalt mixture. It indicates the rubber-modified asphalt overlay with rubber-modified asphalt leveling layer would provide the best cracking resistance compared with the rubber-modified asphalt overlay with conventional asphalt leveling layer and conventional asphalt overlay with rubber-modified asphalt leveling layer. And the conven-tional asphalt overlay with conventional asphalt leveling layer would be expected the worst cracking performance. “
- The sentence has been added into Line 299-300 “The predicted rutting and cracking results are in agreement with the dynamic modulus and fracture properties experimental test results.”
